# The Role of Erythrocyte Sedimentation Rate (ESR) in Myeloproliferative and Lymphoproliferative Diseases: Comparison between DIESSE CUBE 30 TOUCH and Alifax Test 1

**DOI:** 10.3390/diseases11040169

**Published:** 2023-11-15

**Authors:** Martina Pelagalli, Flaminia Tomassetti, Eleonora Nicolai, Alfredo Giovannelli, Silvia Codella, Mariannina Iozzo, Renato Massoud, Roberto Secchi, Adriano Venditti, Massimo Pieri, Sergio Bernardini

**Affiliations:** 1Department of Experimental Medicine, University of Rome “Tor Vergata”, Via Montpellier 1, 00133 Rome, Italy; martina.pelagalli@alumni.uniroma2.eu (M.P.); flaminia.tomassetti@students.uniroma2.eu (F.T.); nicolai@med.uniroma2.it (E.N.); alfredo.giovannelli@students.uniroma2.eu (A.G.); silvia.codella@students.uniroma2.eu (S.C.); massoud@med.uniroma2.it (R.M.); bernards@uniroma2.it (S.B.); 2Department of Laboratory Medicine, “Tor Vergata” University Hospital, Viale Oxford 81, 00133 Rome, Italy; mariannina.iozzo@ptvonline.it; 3Department of Biomedicine and Prevention, Hematology, University Tor Vergata, 00133 Rome, Italy; robsecchi922@gmail.com (R.S.); adriano.venditti@uniroma2.it (A.V.)

**Keywords:** ESR, lymphoproliferative diseases, myeloproliferative diseases, cancer

## Abstract

(1) Background: The erythrocyte sedimentation rate (ESR) is widely diffused in hematology laboratories to monitor inflammatory statuses, response to therapies (such as antibiotics), and oncologic diseases. However, ESR is not a specific diagnostic marker but needs to be contextualized and compared with clinical and other laboratory findings. This study aimed to investigate the performance of two automated instruments, namely the DIESSE CUBE 30 TOUCH (DIESSE, Siena, Italy) and the Alifax Test 1 (Alifax Srl, Polverara, Italy), in comparison with the gold standard, the Westergren method, in lymphoproliferative and myeloproliferative patients. (2) Methods: 97 EDTA samples were selected from the hematology department of Roma Tor Vergata Hospital and analyzed. Statistical analysis was applied. (3) A good correlation between CUBE 30 TOUCH and the gold standard was observed in the overall sample (R^2^ = 0.90), as well as in patients with lymphoproliferative diseases (R^2^ = 0.90) and myeloproliferative diseases (R^2^ = 0.90). The correlation between Test 1 and the gold standard was observed in the overall sample (R^2^ = 0.68), as well as in patients with lymphoproliferative diseases (R^2^ = 0.79) and myeloproliferative diseases (R^2^ = 0.53). (4) Conclusions: The CUBE 30 TOUCH appears to be a more trustworthy tool for evaluating ESR in these pathologies.

## 1. Introduction

Hematological disorders categorized as congenital or acquired include both malignant and benign disorders. These disorders affect various components of the blood, including red blood cells (RBC), platelets (PLT), and white blood cells (WBC) [1], and they are both qualitative [2,3] and quantitative disorders [4]. In particular, in quantitative disorders, the involvement of PLT in the intricate interplay between hemostasis, thrombosis, inflammation, and cancer appears to be multifaced, underscoring its substantial importance in each of these disease processes [5].

The myeloproliferative and lymphoproliferative disorders [6,7,8] represent a heterogeneous group of hematologic neoplasms classically characterized by clonal disorder in hematopoietic stem cells (HSC) [9]. The diagnosis of most hematological malignancies involves molecular testing methods, such as immunophenotypic detection using flow cytometry [10], cytogenetic analysis, and fluorescent in situ hybridization (FISH) [11,12], as well as morphological assessment of peripheral blood or bone marrow. Nevertheless, some non-specific diagnostic markers, such as complete blood count (CBC), erythrocyte sedimentation rate (ESR), and C-reactive protein (CRP), are commonly employed in clinical practice to monitor the growth and progression of hematological neoplasms [13].

In light of these considerations, the non-specific marker ESR has high value when combined with some other inflammation markers for assessing heightened or chronic inflammatory activity [14] Inflammation is often associated with the development and progression of cancer [15]. However, it is important to note that ESR does not measure a single substance but rather a physical process [16]. It reflects the physiological mechanism that results in the formation of erythrocyte rouleaux, a phenomenon characterized by the linear arrangement of RBC resembling a stack of coins, due to an increase in positively charged proteins, commonly present in various inflammatory conditions [14].

The ESR is a valuable diagnostic marker for evaluating inflammation in the body, often associated with conditions such as autoimmune diseases, infections, or cancer [17]. High ESR levels have been notably linked to a poor prognosis in various cancers, including hematological diseases like Hodgkin’s lymphoma. This physiological phenomenon was first observed by Dr. Edmund Faustyn Biernacki in 1897 [18,19]. However, it was not until 1921 that Dr. Westergren established the measurement standards for the ESR test, which are still in use today [20]. Moreover, the Westergren technique is still the reference method recommended by the International Committee for Standardization in Hematology (ICSH), although nowadays it is being supplanted by automated ESR analyzers in clinical routine laboratories. This shift is due to the Westergren method’s inherent slowness, taking approximately 1 h to yield results, and its reliance on operator involvement due to a long turnaround time (TAT) along with sample consumption. Consequently, it is less conducive in aiding clinicians in a timely diagnosis [21].

Some new automated methods can produce ESR results in as little as 5 min [22,23], often referred to as “alternative methods” [24]. Others measure sedimentation more rapidly than the Westergren method, without blood consumption and without requiring operator intervention, thereby reducing biological hazards (called “modified methods”).

Plasmatic proteins play a role in reducing erythrocyte aggregation [21,25]. Therefore, during inflammatory conditions, the rise in plasma proteins promotes the formation of erythrocyte rouleaux [20], which settle more readily than individual RBC due to increased aggregation [18]. Qualitative and quantitative alterations in RBC, such as anemia, macrocytosis and microcytosis, can lead to elevated sedimentation rates. Additionally, the presence of abnormal red cell shapes and reduced deformability can impact the sedimentation rate [25,26].

Moreover, in patients with RBC disorders, the interpretation of ESR is influenced by their hematocrit status [27]. Chronic inflammation has been shown to enhance a mutagenic microenvironment that could lead to cancer development. An elevated ESR may serve as an indicator of occult cancer. An association exists between chronic inflammation and carcinogenesis, and subclinical or even undetectable inflammation can be as significant as chronic inflammation in increasing cancer risk, development, and progression [28,29]. Circumstances leading to a decreased ESR can often result in missed diagnoses. Previous studies have identified ESR, although non-specific, as a prognostic factor in patients with hematological malignancies, such as Hodgkin’s disease [29,30], where it also plays a role in predicting early disease relapse [31,32].

Moreover, an increased ESR frequently indicates the progression of these underlying mechanisms and can serve as a valuable prognostic factor [33], often when combined with the RBC count [34]. Recent findings have challenged the hypothesis that patients with myeloproliferative neoplasms are at an increased risk of developing lymphoproliferative malignancies [35]. Furthermore, elevated ESR, CRP, and lactate dehydrogenase (LDH) levels have been identified as adverse prognostic factors impacting the survival of cancer patients [36,37,38,39].

In light of these considerations, this study aims to investigate the performance of two automat ESR analyzers based on different methodologies, namely CUBE 30 TOUCH (DIESSE, Siena, Italy) and Test 1 (Alifax Srl, Polverara, Italy), using samples from the Department of Lymphoproliferative and Myeloproliferative Diseases. The performance of these analyzers is compared with that of the gold standard method, the Westergren method.

## 2. Materials and Methods

### 2.1. Samples

In total, 97 blood samples preserved in EDTA were collected, comprising 37 from females and 60 from males, with an average age of 63.7 ± 12.4 years, ranging from 28 to 83 years. These samples were obtained from routine leftover tubes originating from patients at Tor Vergata University Hospital and were retrieved by the hematology department. All samples were anonymized to protect patients’ privacy. The patient cohort was categorized based on the etiology of their diseases into two main groups: hematological oncologic disorders (N = 88) and non-oncologic disorders (N = 9). Hematologic oncologic disorders included patients afflicted with myeloproliferative diseases (N = 38) and lymphoproliferative disease (N = 50). The myeloid diseases included cases of acute myeloid leukemia (AML) (N = 37) and myelodysplastic syndromes (MDS) (N = 1). Lymphoproliferative diseases comprised acute lymphoid leukemia (ALL) (N = 12), lymphoma (N = 25), and multiple myeloma (MM) (N = 13). Non-oncologic hematologic disorders included thrombotic thrombocytopenic purpura (TTP) (N = 5), reactive lymphadenitis (N = 2), and hematopoietic stem cell donors (CSE donors) (N = 1).

The samples were analyzed within 4 h of blood collection and stored at a refrigerated temperature of +4 °C during the analysis phase, with a maximum storage duration of 3 h. For this study, leftover samples from routine laboratory procedures were utilized. The laboratory’s routine ESR analyzer was Test 1 and all ESR samples were analyzed using this instrument. Samples that exhibited alterations, such as hemolysis, lipemia, and jaundice, were excluded from the study to ensure that their presence would not affect the results.

The study received approval from the Ethics Committee of the “Tor Vergata” Hospital (R.S.202/19) and was conducted in accordance with the revised Declaration of Helsinki. Informed consent was obtained from all subjects who participated in the study.

### 2.2. DIESSE CUBE 30 TOUCH

The DIESSE CUBE 30 TOUCH (DIESSE, Siena, Italy) is an automated ESR analyzer, classified under the “Modified Westergren method” [24]. The CUBE 30 TOUCH is compatible with standard 13 × 75 mm K2EDTA and K3EDTA tubes, ranging from 1.0 to 4.0 mL in volume [40]. This analyzer operates without the need for reagents, eliminating the requirement for operator exposure to patient samples. The instrument performs direct testing from EDTA tubes without any handling or consumption of patient samples. Results are processed within 20 min, allowing for testing of up to 90 samples per hour.

Quality controls are processed prior to each analytical measurement and are automatically stored by the instrument.

### 2.3. Alifax Test 1

Alifax Test 1 (Alifax Srl, Polverara, Italy) is an automated ESR analyzer, classified as an “Alternate Westergren method” [6]. Test 1 employs capillary photometric kinetic technology. Erythrocyte sedimentation is monitored using a photometer and the results are reported in millimeters per hour (mm/h). Results are available within 20 s for a single sample and approximately 20 min for a batch of 40 samples. Similar to the previously mentioned analyzer, Test 1 eliminates the need for operator exposure to patient samples, as it pierces the rubber cap of the tube with a fixed needle directly through the cap, with processed samples being disposed of in a waste tank. However, this technology does consume a portion of the sample.

Quality controls were processed each morning, following the laboratory’s routine protocol.

### 2.4. Westergren Method

A total of 1 mL of venous blood (K3EDTA) is collected manually via pipettes into a full line of Westergren in a tube containing 0.250 mL of sodium citrate. After being mixed gently, blood is drawn into the standardized Westergren tube, marked to 200 mm. This tube is then placed vertically in a rack at room temperature for 1 h. The visual determination of the result was identified by the mark where the upper limit of erythrocyte sedimentation was sedimented due to gravity. The Westergren method was performed manually according to the ICSH’s recommendations, within 4 h of blood sampling [20].

### 2.5. Statistics

Initially, to assess the equality between two one-dimensional probability distributions, Kolmogorov–Smirnov tests with a confidence interval (CI) of 95% were performed. Once the dataset was established to be linearly related and bivariant distributed, Passing–Bablok correlation analysis was evaluated to compare the two new methods to the Westergren test, using the nonparametric measure of rank correlation (statistical dependence between the rankings of two variables), and the Spearman rank correlation coefficient (R^2^). Passing–Bablok regression and Spearman’s ρ were used for an overview of the pairwise method comparisons.

Lastly, the Bland–Altman test was performed as an alternative analysis to compare two clinical measurements, each of which produced some error in their measures, and to determine the agreement between the methods by studying the mean difference and constructing limits of agreement.

All data were examined using Med Calc Ver.18.2.18 (MedCalc Software Ltd., Ostend, Belgium).

The statistical significance level established for all tests performed was *p* < 0.05.

## 3. Results

A total of 97 samples were collected from units treating patients with myeloproliferative and lymphoproliferative diseases at the Tor Vergata Hospital in Rome and these samples were analyzed for ESR determination. The study population allowed for the classification of the samples into two main groups: samples from patients with malignant disorders and samples from patients with benign disorders. The specific details of the study population are provided in Table 1. Samples from patients with benign hematological disorders, due to their limited number (N = 8), were included only in the statistical analysis conducted on the entire sample set.

The samples were analyzed using both methods and the results were compared. The one-dimensional distributions were assessed using the Kolmogorov–Smirnov test, and, consequently, the Spearman rank correlation coefficient (R^2^) was employed. The Spearman rank correlation coefficient ranges from +1 to −1, where a value of +1 indicates a perfect positive association, and −1 indicates a perfect negative association.

### 3.1. Overall Samples

Figure 1 illustrates the Passing–Bablok analysis for the overall samples between CUBE 30 TOUCH versus the Westergren method (A) and between Test 1 and Westergren (B). Figure 1A shows an excellent correlation between the CUBE 30 TOUCH and Westergren method with the equation of y = −0.78 + 1.48 × (Intercept A: −0.79, 95% CI: −3.10 to 1.21; Slope B: 1.48, 95% CI: 1.23 to 1.63) and Spearman coefficient (R^2^) of 0.90 (95% CI: 0.86 to 0.93; *p* < 0.001); Figure 1B shows a moderate correlation between the Test 1 and Westergren method with equation of y = 0.60 + 0.70 × (Intercept A: −0.60, 95% CI: −1.86 to 1.79; Slope B: 0.70, 95% CI: 0.60 to 0.83) and Spearman coefficient (R^2^) of 0.68 (95% CI: 0.60 to 0.77; *p* < 0.001).

### 3.2. Samples from Patients with Lymphoproliferative and Myeloproliferative Diseases

Figure 2 shows the correlation of CUBE 30 TOUCH versus Westergren (A − C) and between Test 1 and Westergren (B − D) of data from the patients divided into lymphoproliferative (A − B) and myeloproliferative diseases (C − D). Figure 2A shows a high correlation between the CUBE 30 TOUCH and the Westergren method in lymphoproliferative disease patients with an equation of y = −2.66 + 1.49 × (Intercept A: −2.66, 95% CI: −8.82 to 0.65; Slope B: 1.49, 95% CI: 1.07 to 1.91) and Spearman coefficient (R^2^) of 0.90 (95% CI: 0.82 to 0.94; *p* < 0.001); Figure 2B shows a good agreement between Test 1 and the Westergren method in lymphoproliferative disease patients with an equation of y = −1.70 + 0.08 × (Intercept A: −1.70. 95% CI: −3,72 to 3.34; Slope B: 0.80, 95% CI: 0.65 to 1.11) and Spearman coefficient (R^2^) of 0.79 (95% CI: 0.66 to 0.88; *p* < 0.001). Figure 2C shows the accordance between the CUBE 30 TOUCH and the Westergren method in the myeloproliferative disease patients with an equation of y = 1.13 + 1.39 × (Intercept A: 1.13, 95% CI: −3.00 to 6.28; Slope B: 1.39, 95% CI: 1.00 to 1.64) and Spearman coefficient (R^2^) of 0.90 (95% CI: 0.82 to 0.94; *p* < 0.001); Figure 2D shows the correlation between Test 1 and the Westergren method in myeloproliferative disease patients with an equation of y = 0.76 + 0.56 × (Intercept A: 0.76, 95% CI: −8.82 to 0.65; Slope B: 0.56, 95% CI: 1.07 to 1.91) and Spearman coefficient (R^2^) of 0.53 (95% CI: 0.25 to 0.73; *p* < 0.001).

### 3.3. Samples from Patients at Onset Symptoms and under Pharmaceutical Treatment Patients

Figure 3 illustrates the correlation of CUBE 30 TOUCH versus the Westergren procedure (A−C) and between Test 1 and Westergren (B−D) of data from the patients divided into onset (A−B) and under pharmaceutical treatment (C−D) patients. Figure 3A shows a high correlation between the CUBE 30 TOUCH and Westergren method for the disease-onset patients with an equation of y = 1.67 + 1. 33 × (Intercept A: 1.67, 95% CI: −3.77 to 6,82; Slope B: 1.33, 95% CI: 0.88 to 1.74) and Spearman coefficient (R^2^) of 0.84 (95% CI: 0.72 to 0.91; *p* < 0.001); Figure 3B shows a low correlation between Test 1 and the Westergren method for disease-onset patients with an equation of y = 0.67 + 0.67 × (Intercept A: 0.67, 95% CI: −3.93 to 2.78; Slope B: 0.67, 95% CI: 0.50 to 1.00) and Spearman coefficient (R^2^) of 0.50 (95% CI: 0.24 to 0.69; *p* < 0.001). Figure 3C shows a high correlation between the CUBE 30 TOUCH and the Westergren method in patients under pharmaceutical treatment with an equation of y = −1.19 + 1.52 × (Intercept A: −1.19, 95% CI: −5,67 to 0.47; Slope B: 1.52, 95% CI: 1.23 to 1.75) and Spearman coefficient (R^2^) of 0.92 (95% CI: 0.86 to 0.95; *p* < 0.001); Figure 3D shows a good correlation between Test 1 and the Westergren method in patients under pharmaceutical treatment with an equation of y = 0.90 + 0.70 × (Intercept A: 0.90, 95% CI: −2.03 to 5,33; Slope B: 0.70, 95% CI: 0.59 to 0.84) and a Spearman coefficient (R^2^) of 0.79 (95% CI: 0.65 to 0.87; *p* < 0.001).

## 4. Discussion

The ESR is widely used in laboratory routines, though limited information is present to discriminate between pathological states (such as inflammatory diseases and malignancies) and non-pathological states in general medical practice [41] Increasing evidence suggests that the majority of tumors are associated with chronic inflammation, leading to common laboratory abnormalities that include neutrophilic leukocytosis, liver biomarkers, and elevated acute-phase reactants, including the ESR [18,42,43].

However, other research indicates that the ESR can also serve as a tool for long-term monitoring. This emphasizes the need for regular evaluation of these patients at a low healthcare cost [44,45]. Hence, the objective of this study was to assess the performance of two automated instruments, namely the DIESSE CUBE 30 TOUCH (DIESSE, Siena, Italy) and the Alifax Test 1 (Alifax Srl, Polverara, Italy), in comparison to the gold standard, the Westergren method. The adoption of these novel ESR methods facilitates method standardization, laboratory automation, increased work efficacy, reduced costs (as the CUBE 30 TOUCH is a waste free instrument), and ensures the safety of employees by eliminating the need for handling blood [46]. The CUBE 30 TOUCH instrument exhibited a higher level of agreement with the Westergren method when measuring ESR, both across all samples and within the patient groups with lymphoma and myeloproliferative diseases. We observed a strong correlation between the results obtained from the overall sample set and the Westergren method, with a Spearman coefficient (R^2^) of 0.90 (Figure 1A). The Spearman coefficient between Test 1 and the Westergren method was 0.68 (Figure 1B). Due to the limited number of patients in the hematological benign disorders group, a comparative analysis was not feasible. As a result, our subsequent step involved stratifying the study population into two groups, those with myeloproliferative diseases and those with lymphoproliferative diseases, to evaluate the correlation between CUBE 30 TOUCH and the Westergren method, as well as between Test 1 and the Westergren method (Figure 2). In patients with lymphoproliferative disease, both the CUBE 30 TOUCH and Test 1 devices exhibit a strong correlation with the Westergren method. The Spearman coefficient for CUBE 30 TOUCH versus Westergren was 0.90, while for Test 1 versus Westergren it was 0.79. Conversely, in patients with myeloproliferative diseases, there was a Spearman coefficient of 0.90 for CUBE 30 TOUCH, whereas Test 1 showed a moderate correlation with a Spearman coefficient of 0.53 compared to the Westergren method (Figure 2).

The well-established knowledge that cancer conditions are associated with chronic inflammatory status supports the role of the ESR test as the most commonly used and suitable marker for tracking inflammation.

However, many pathways of the inflammatory process are interconnected with tumorigenesis, development, interaction, and therapy [29,43]. In the management of cancer, a high ESR is associated with an unfavorable prognosis in malignancies due to an increase in serum proteins, such as CRP, in response to inflammation, infection, and malignancy [47,48,49]. The prognostic value of ESR changes independently of other prognostic factors and treatments. This is in contrast to CRP, which increases only at the onset of inflammation and the WBC count, including the neutrophil-to-lymphocyte ratio (NLR), which can exhibit daily fluctuations in patients undergoing pharmaceutical therapy [50].

Elevated ESR levels have been correlated with an increased incidence of early relapse in cancer patients [32,49]. It has been observed that an elevated ESR value is linked to higher mortality rates in patients with hematological disorders, suggesting the presence of persistent inflammatory status. Therefore, our study aimed to assess the performance of two automated ESR analyzers in comparison to the gold standard in patients with hematological disorders at the disease’s onset and during treatment. We found a stronger correlation between the CUBE 30 TOUCH and the Westergren method, compared to TEST 1, at disease onset, with respective Spearman coefficients of 0.84 and 0.50. A similar trend was observed in samples from patients undergoing treatment (Spearman’s coefficient 0.92 for CUBE 30 TOUCH versus Westergren and 0.79 for Test1 vs. Westergren). These results highlight the ability of CUBE 30 TOUCH to accurately detect dynamic changes in ESR. This is valuable, especially when considered alongside other blood inflammatory biomarkers, for assessing treatment effectiveness, predicting relapses, and ensuring reliability in critical pathologies where a decreased ESR value could result in missed or delayed diagnoses [49]. Furthermore, fluctuations in these inflammatory parameters lead to changes in serum protein levels, thus proportionally affecting the ESR process. Nonetheless, the inflammatory activities within the tumor microenvironment that lead to an increased ESR value still require further clarification, despite the assumption of its correlation with tumor burden, histologic grade, and anemia [50].

Moreover, our findings demonstrate the excellent performance of CUBE 30 TOUCH in lymphoproliferative and myeloproliferative diseases, suggesting that ESR could serve as a cost-effective biomarker for risk assessment in patients with lymphoproliferative and myeloproliferative disease. This also indicates a superior performance when using the modified method compared to an alternative one. Moreover, the CUBE 30 TOUCH showed no interference with low hematocrit value and low hemoglobin concentration [51]. Furthermore, the DIESSE analyzer, based on an optical method, does not require blood consumption resulting in zero waste production, reducing the biological hazard and increasing the compliance of the laboratory operator. Such information can assist clinicians in identifying and customizing suitable treatments and predicting relapse in specific hematological settings of patients, where ESR plays a recognized role.

### Limitations of the Study

Despite the numerous advantages of the new CUBE 30 TOUCH and the positive findings of this study, some limitations were registered. Firstly, the number of hematological patients should be increased. Increasing the population size would enable the evaluation of the correlation of this parameter in various types of leukemias or lymphomas. It is well established that in some of these conditions, the inflammatory role predominates over others (for example in Hodgkin’s lymphoma). Second, different pharmacological treatments were not captured in our study, which might lead to some altered ESR evaluation. For instance, it would be important to assess statistical significance when using of anti-inflammatory therapies (such as steroids), which are often linked to chemotherapy regimens for lymphoproliferative syndromes. Therefore, further studies are required to increase the numerosity of our cohort and to confirm the optimal correlation between CUBE 30 TOUCH and the Westergren method in hematological patients.

## Figures and Tables

**Figure 1 diseases-11-00169-f001:**
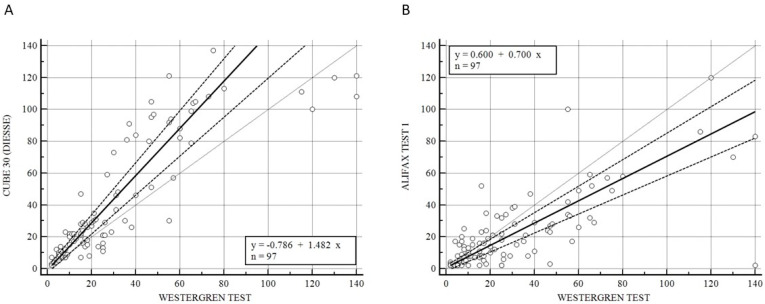
(**A**) Passing–Bablok correlation for the overall samples between CUBE 30 TOUCH versus Westergren method. (**B**) Passing–Bablok correlation for the overall samples between Test 1 versus Westergren method. [Solid line: regression line; dashed lines: the confidence interval’s lines; circles: identity line (x = y)].

**Figure 2 diseases-11-00169-f002:**
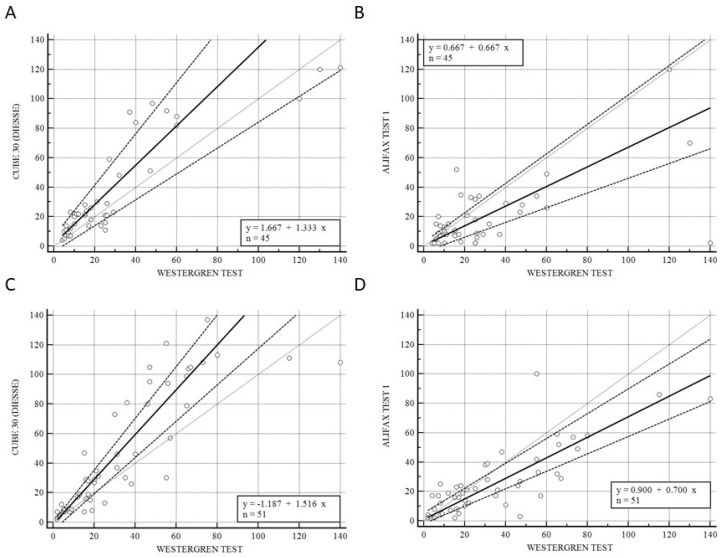
(**A**) Passing–Bablok correlation of CUBE 30 TOUCH versus Westergren method of data from the patients into lymphoproliferative diseases. (**B**) Passing–Bablok correlation of Test 1 versus Westergren method of data from the patients into lymphoproliferative diseases. (**C**) Passing–Bablok correlation of CUBE 30 TOUCH versus Westergren method of data from the patients into myeloproliferative diseases. (**D**) Passing–Bablok correlation of Test 1 versus Westergren method of data from the patients into myeloproliferative diseases. [Solid line: regression line; dashed lines: the confidence interval’s lines; circles: identity line (x = y)].

**Figure 3 diseases-11-00169-f003:**
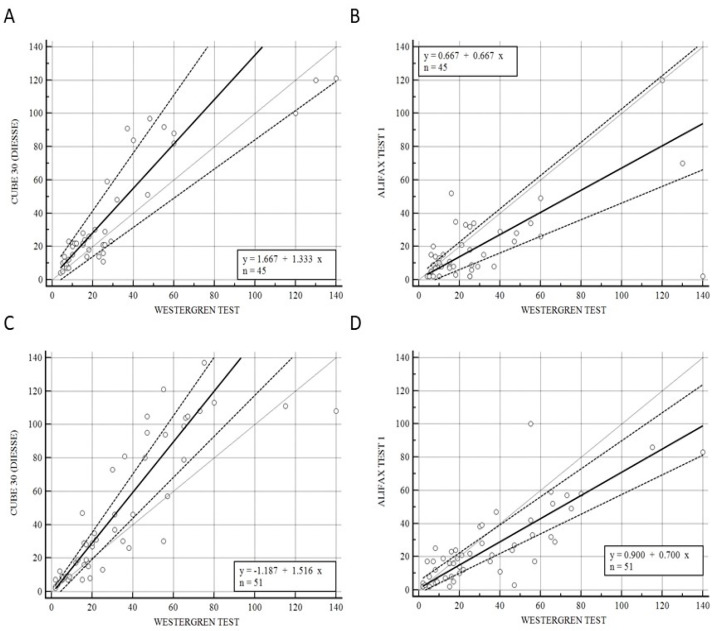
(**A**) Passing–Bablok correlation of CUBE 30 TOUCH versus Westergren method of data from patients at the onset of diseases. (**B**) Passing–Bablok correlation of Test 1 versus Westergren method of data from patients at the onset of diseases. (**C**) Passing–Bablok correlation of CUBE 30 TOUCH versus Westergren method of data from patients under pharmaceutical treatment. (**D**) Passing-Bablok correlation of Test 1 versus Westergren method of data from patients under pharmaceutical treatment. [Solid line: regression line; dashed lines: the confidence interval’s lines; circles: identity line (x = y)].

**Table 1 diseases-11-00169-t001:** Diagnostic characterization of the population: 88 samples from hematological malignant disorders (both myeloproliferative and lymphoproliferative); 37 samples with AML, 1 sample with MDS, 12 samples with ALL, 25 samples with lymphoma, and 15 samples with multiple myeloma; 8 samples from hematological benign disorders; 5 samples with TTP, 2 samples with RA, and 1 CDE sample.

Type of Disorder	Group of Diseases	Type of Disease	Subtype of Disease
HAEMATOLOGICAL MALIGNANT DISORDERS (N = 88)	Myeloproliferative diseases (N = 38)	Acute myeloid leukemia (AML) (N = 37)	Acute myeloid leukemia (non-APL) (N = 35) Acute promyelocytic leukemia (APL) (N = 2)
		Myelodysplastic syndromes (MDS) (N = 1)	
	Lymphoproliferative diseases (N = 50)	Acute lymphoid leukemia (ALL) (N = 12)	Lymphoblastic leukemia B (ALL-B) (N = 5) Lymphoblastic leukemia T (ALL-T) (N = 2) Chronic Lymphatic Leukemia (CLL) (N = 1) Acute Lymphoblastic Leukemia (ALL) (N = 4)
		Lymphoma (N = 25)	Diffuse large cell B lymphoma (N = 4) Richter syndrome (N = 2) Lymphoblastic Lymphoma (N = 6) Burkitt’s lymphoma (N = 4) Hodgkin’s lymphoma (N = 6) Non-Hodgkin’s lymphoma (N = 3)
		Multiple myeloma (MM) (N = 13)	
HAEMATOLOGICAL BENIGN DISORDERS (N = 8)		Thrombotic thrombocytopenic purpura (TTP) (N = 5)	
		Reactive lymphadenitis (RA) (N = 2)	
		Hematopoietic stem cell donors (CSE) (N = 1)	

## Data Availability

The data that support the findings of this study are available from the corresponding author upon reasonable request.

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
