# Peer review of "The Role of Erythrocyte Sedimentation Rate (ESR) in Myeloproliferative and Lymphoproliferative Diseases: Comparison between DIESSE CUBE 30 TOUCH and Alifax Test 1"

_diseases, 2023, doi:10.3390/diseases11040169_

Round 1
Reviewer 1 Report
Comments and Suggestions for Authors
Dear Sir
I carefully read the manuscript by Pelagalli M and coworkers.
“The role of Erythrocyte Sedimentation Rate (ESR) in myeloproliferative and lymphoproliferative diseases: comparison between DIESSE CUBE 30 TOUCH and Alifax Test 1”
Manuscript ID: diseases-2561933
The aim of the study is to evaluate ESR. in a group of 97 patients with myeloproliferative or lymphoproliferative disease ESR evaluation was performed using two commercial methods: Diesse Cube 30 Touch and Alifax Test 1, in relation to the classic method of Westergren (which is still the ICSH reference method). It is therefore a simple comparative study between analytical methods made interesting by the selection of patients.
The study design appears to be correct.
The statistical processing is adequate.
The English language needs proper revision.
Unfortunately, the text is extremely verbose.
The introduction is 94 lines long, I believe it could be condensed into no more than 20-25 lines without losing anything in clarity and remaining perfectly functional for the purpose. For example, lines 33 to 59 cover myeloproliferative and lymphoproliferative diseases, this aspect does not seem relevant to the study. From line 60 to line 82 general aspects concerning the ESR are dealt with, also in this case these aspects do not appear to be relevant. Also, from line 99 at the end of the introduction the material should be eliminated or condensed into 1-2 short sentences.
Materials and Methods is equally verbose. The selection of patients is reported in a table, it is not necessary to repeat it in the text. The authors compare the performance of three routine commercial methods, it is not necessary to describe in detail the analytical methodologies used.
The results section appears verbose but essentially correct.
Discussion occupies lines 290 to 364. It should be drastically shortened.
The work is deemed unacceptable in this version and should be rejected. Possible an ex novo evaluation of a widely revised version.
Comments on the Quality of English Language
The English language needs proper revision.
Author Response
Reviewer 1
The introduction is 94 lines long, I believe it could be condensed into no more than 20-25 lines without losing anything in clarity and remaining perfectly functional for the purpose. For example, lines 33 to 59 cover myeloproliferative and lymphoproliferative diseases, this aspect does not seem relevant to the study. From line 60 to line 82 general aspects concerning the ESR are dealt with, also in this case these aspects do not appear to be relevant. Also, from line 99 at the end of the introduction the material should be eliminated or condensed into 1-2 short sentences.
R: Following the Reviewer's suggestion, we summarized the introduction paragraph.
Materials and Methods is equally verbose. The selection of patients is reported in a table, it is not necessary to repeat it in the text. The authors compare the performance of three routine commercial methods, it is not necessary to describe in detail the analytical methodologies used.
R: The part referring to the selection of patients was rewritten according to the other Reviewers’ suggestions, and Table 1 was relocated in the Results. Some information in the description of the three routine commercial methods was deleted.
The results section appears verbose but essentially correct.
R: The English has been revised
Discussion occupies lines 290 to 364. It should be drastically shortened.
R: The Discussion was summarized and rephrased to be more incisive and focused.

Reviewer 2 Report
Comments and Suggestions for Authors
The manuscript titled “The role of Erythrocyte Sedimentation Rate (ESR) in myeloproliferative and lymphoproliferative diseases: comparison between DIESSE CUBE 30 TOUCH and Alifax Test 1” explores the correlation between two innovative methods for assessing ESR, in contrast to the conventional Westergren method, in both healthy individuals and those with myeloproliferative and lymphoproliferative diseases. While the manuscript presents an intriguing perspective with potential implications for clinical practice, the authors are encouraged to address the following comments before proceeding further.
Major comments
1) In the "Introduction" section, the authors should focus on the central theme of the paper. For instance, there is no need to delve into an extensive discussion about the types of white blood cells found in the blood or all the diseases characterized by their abnormalities. Although the historical context surrounding ESR is captivating, it is not the primary focus of the study and could be omitted. Consequently, the "Introduction" section requires revision and condensation.
2) It is recommended that the description of Table 1 be relocated to the "Results" section rather than the "Methods" section.
3) As indicated by the authors in the "Introduction" section, hematocrit and hematimetric indexes can significantly influence ESR results. Hence, it is imperative for the authors to address these parameters when comparing different groups.
4) Please include demographic data such as age, sex, and other relevant factors in the manuscript
5) It is advisable to incorporate a dedicated section addressing the limitations of the study.
6) There is room for improvement in terms of English grammar and style throughout the manuscript.
Comments on the Quality of English LanguageThere is room for improvement in terms of English grammar and style throughout the manuscript.
Author Response
Reviewer 2
The manuscript titled “The role of Erythrocyte Sedimentation Rate (ESR) in myeloproliferative and lymphoproliferative diseases: comparison between DIESSE CUBE 30 TOUCH and Alifax Test 1” explores the correlation between two innovative methods for assessing ESR, in contrast to the conventional Westergren method, in both healthy individuals and those with myeloproliferative and lymphoproliferative diseases. While the manuscript presents an intriguing perspective with potential implications for clinical practice, the authors are encouraged to address the following comments before proceeding further.
Major comments
- In the "Introduction" section, the authors should focus on the central theme of the paper. For instance, there is no need to delve into an extensive discussion about the types of white blood cells found in the blood or all the diseases characterized by their abnormalities. Although the historical context surrounding ESR is captivating, it is not the primary focus of the study and could be omitted. Consequently, the "Introduction" section requires revision and condensation.
R: Following the reviewer's suggestion, we summarized the introduction paragraph.
2) It is recommended that the description of Table 1 be relocated to the "Results" section rather than the "Methods" section.
R: Thanks for pointing this out, we relocated Table 1, following the suggestion.
Minor comments
3) As indicated by the authors in the "Introduction" section, hematocrit and hematimetric indexes can significantly influence ESR results. Hence, it is imperative for the authors to address these parameters when comparing different groups.
R: Our study was focused on instrument validation, aiming to compare the measure obtained by the automated method with the gold standard in particular patients with pathological hematological diseases. However, the hematocrit study on the possible interference in the ESR values was previously published in another work of our team. We added a sentence in the text and the relate reference [53].
4) Please include demographic data such as age, sex, and other relevant factors in the manuscript
R: We included in the Material and methods, the missing data.
5) It is advisable to incorporate a dedicated section addressing the limitations of the study.
R: We included at the end of the Discussion the limitations of this study.
6) There is room for improvement in terms of English grammar and style throughout the manuscript.
R: English has been revised throughout the manuscript.
Comments on the Quality of English Language
- There is room for improvement in terms of English grammar and style throughout the manuscript.
R: English has been revised throughout the manuscript.

Reviewer 3 Report
Comments and Suggestions for Authors
The paper evaluates and underscores the superior accuracy of the DIESSE CUBE 30 TOUCH instrument over the Alifax Test 1 in measuring the Erythrocyte Sedimentation Rate (ESR) in myeloproliferative and lymphoproliferative diseases, emphasizing its potential for enhanced diagnostic precision in these pathologies."
Major issues
1. Line 221-223: The definition of Spearman Rank Correlation coefficient is wrong, it is oversimplified and partially inaccurate. "-1" doesn't mean "no association", but rather a perfect negative association. Suggestion: "The Spearman Rank Correlation coefficient ranges from +1 to -1, where a value of +1 indicates a perfect positive association, and -1 indicates a perfect negative association."
2. In the material and methos the authors said that they used Pasinb Bavlok regression, but in the results the say that they us Passing bablok correlation. CLine 237-239, 259-264, 284-289: Regression is not the same that correlation. Consistency in naming the correlations is needed. If it's introduced as "Passing-Bablok correlation", then maintain that. Alternatively, if you're using linear regression, then "Passing-Bablok" might not be the right term to use.
MINOR ISSUES
Review of the Material and Methods section:
3. Line 130-141: The order of introducing the samples is confusing. It may be more effective to first give the total sample number, then the breakdown into different diseases, and then the specific numbers for each disease.
4. Table 1:The categorization seems repetitive. E.g., Acute myeloid leukemia (AML) is mentioned twice. Is that correct?
5. Redundancy: Line 147: "Only the samples coming from the Hematology department’s patients were also processed with the new analyzer CUBE 30 TOUCH." This is redundant since it was mentioned earlier that the samples were from the Hematology department.
6. Line 143-148: The specific duration the samples were stored at +4°C before analysis should be specified.
7. Instrument Details: Line 164-172: It might be useful to explain why these particular features of the CUBE 30 TOUCH are relevant to this study, as there's a lot of detailed technical information that may or may not be crucial to the study's context.
8. To introduce photos of the two devices may improve the paper providing the reader and idea about the machines.
9. Line 203-215: provide a brief rationale for the choice of each statistical method to offer readers insight into why these methods were the best choices for the data analysis.
10. Ethical Consideration: Line 159-161: It would be helpful to specify if the leftover blood samples were anonymized and if any personal patient information was accessed during the study.
Comments on the Quality of English LanguageThere are many Typos and Grammar errors:
Line 145: "all the ESR samples" might be better as "all the samples for ESR".
Line 172: "test up to 90 samples" might be clearer as "tests up to 90 samples".
Line 187: The word "Anyhow" seems informal. Consider replacing it with "However".
Line 197: "sediment was sedimented" - the wording is redundant.
Line 153: "form" should be corrected to "from".
Line 217-218: The phrasing is slightly awkward. It should be rephrased for clarity. Suggestion: "A total of 97 samples were collected from units treating myeloproliferative and lymphoproliferative diseases at the Hospital of Tor Vergata, Rome."
Author Response
Reviewer 3
The paper evaluates and underscores the superior accuracy of the DIESSE CUBE 30 TOUCH instrument over the Alifax Test 1 in measuring the Erythrocyte Sedimentation Rate (ESR) in myeloproliferative and lymphoproliferative diseases, emphasizing its potential for enhanced diagnostic precision in these pathologies."
Major issues
- Line 221-223: The definition of Spearman Rank Correlation coefficient is wrong; it is oversimplified and partially inaccurate. "-1" doesn't mean "no association", but rather a perfect negative association. Suggestion: "The Spearman Rank Correlation coefficient ranges from +1 to -1, where a value of +1 indicates a perfect positive association, and -1 indicates a perfect negative association."
R: Thanks for pointing this out, we corrected the imprecision.
- In the material and methos the authors said that they used Pasinb Bavlok regression, but in the results the say that they us Passing bablok correlation. CLine 237-239, 259-264, 284-289: Regression is not the same that correlation. Consistency in naming the correlations is needed. If it's introduced as "Passing-Bablok correlation", then maintain that. Alternatively, if you're using linear regression, then "Passing-Bablok" might not be the right term to use.
R: We are sorry for the errors, we corrected them throughout the text.
Minor issues
Review of the Material and Methods section:
- Line 130-141: The order of introducing the samples is confusing. It may be more effective to first give the total sample number, then the breakdown into different diseases, and then the specific numbers for each disease.
R: We rephrased the paragraph.
- Table 1: The categorization seems repetitive. E.g., Acute myeloid leukemia (AML) is mentioned twice. Is that correct?
R: Thanks for pointing this out, we used the term AML for grouping all the acute myeloid pathologies and we introduced the differentiation for APL AML and non-APL AML to differentiate promyelocytic leukemias from myeloid leukemias.
- Redundancy: Line 147: "Only the samples coming from the Hematology department’s patients were also processed with the new analyzer CUBE 30 TOUCH." This is redundant since it was mentioned earlier that the samples were from the Hematology department.
R: We rephrased the sentence, following the Reviewer’s suggestion.
- Line 143-148: The specific duration the samples were stored at +4°C before analysis should be specified.
R: We included the missing information in the Material and methods.
- Instrument Details: Line 164-172: It might be useful to explain why these particular features of the CUBE 30 TOUCH are relevant to this study, as there's a lot of detailed technical information that may or may not be crucial to the study's context.
R: We commented on this information in the Discussion.
- To introduce photos of the two devices may improve the paper providing the reader and idea about the machines.
R: In order to avoid unnecessarily increasing the workload, we have chosen not to include photographs of the instruments, which are nevertheless available on the company's website.
- Line 203-215: provide a brief rationale for the choice of each statistical method to offer readers insight into why these methods were the best choices for the data analysis.
R: We explain the statistical choice in the Material and methods.
- Ethical Consideration: Line 159-161: It would be helpful to specify if the leftover blood samples were anonymized and if any personal patient information was accessed during the study.
R: Samples were anonymized, and a sentence has been added in the material and method
Comments on the Quality of English Language
There are many Typos and Grammar errors:
Line 145: "all the ESR samples" might be better as "all the samples for ESR".
R: We corrected the error.
Line 172: "test up to 90 samples" might be clearer as "tests up to 90 samples".
R: We corrected the error.
Line 187: The word "Anyhow" seems informal. Consider replacing it with "However".
R: We corrected the error.
Line 197: "sediment was sedimented" - the wording is redundant.
R: We corrected the error.
Line 153: "form" should be corrected to "from".
R: We corrected the error.
Line 217-218: The phrasing is slightly awkward. It should be rephrased for clarity. Suggestion: "A total of 97 samples were collected from units treating myeloproliferative and lymphoproliferative diseases at the Hospital of Tor Vergata, Rome."
R: We rephrased the sentence.

Round 2
Reviewer 1 Report
Comments and Suggestions for Authors
I found the manuscript still rather verbose, so I took the liberty of proceeding directly to highlight in the file (red strikethrough text) the sentences that in my opinion should be eliminated

Author Response
Dear Reviewer,
On behalf of the co-authors, I am grateful for the constructive feedback. While I acknowledge your recommendations to eliminate certain detailed sentences in the Introduction and Discussion, we believe that some of these hold significance for the context and reinforce the paper's central aim. Moreover, some of the sentences with red strikethrough were written in response to the other Reviewers' concerns during the first revision of the manuscript.
Furthermore, the article must meet a minimum of 4000-word count to address editorial requirements.
For these reasons, we deleted just some of the indicated sentences.
Reviewer 2 Report
Comments and Suggestions for Authors
No further comments. Congratulations on your study.
Author Response
Thank you for your comments